# The Utility of Oral Vitamin B1 and Mecobalamin to Improve Corneal Nerves in Dry Eye Disease: An In Vivo Confocal Microscopy Study

**DOI:** 10.3390/nu14183750

**Published:** 2022-09-11

**Authors:** Xiaotong Ren, Yilin Chou, Yuexin Wang, Dalan Jing, Yanyan Chen, Xuemin Li

**Affiliations:** 1Department of Ophthalmology, Peking University Third Hospital, Beijing 100191, China; 2Department of Ophthalmology, BenQ Medical Center, The Affiliated BenQ Hospital of Nanjing Medical University, Nanjing 210000, China; 3Department of Ophthalmology, Daqing Oilfield General Hospital, Daqing 163311, China

**Keywords:** corneal nerve parameters, dry eye disease (DED), in vivo confocal microscopy (IVCM), mecobalamin, vitamin B1

## Abstract

Our purpose is to demonstrate the changes in cornea nerve parameters and symptoms and signs in dry eye disease (DED) patients after oral vitamin B1 and mecobalamin treatment. In this randomized double-blind controlled trial, DED patients were randomly assigned to either the treatment group (oral vitamin B1 and mecobalamin, artificial tears) or the control group (artificial tears). Corneal nerve parameters via in vivo confocal microscopy (IVCM), DED symptoms, and signs were assessed at baseline and 1 and 3 months post-treatment. In total, 398 eyes from 199 patients were included. In the treatment group, there were significant improvements in corneal nerve length, width, and neuromas, the sign of conjunctival congestion score (CCS), symptoms of dryness, pain, photophobia, blurred vision, total symptom score, and OSDI (OSDI) at 1/3 months post-treatment (all *p* < 0.05). Patients who received vitamin B1 and mecobalamin showed greater improvement in CCS, dryness scores at 1 month (*p* < 0.05), corneal fluorescein staining (CFS) (*p* = 0.012), photophobia (*p* = 0.032), total symptom scores (*p* = 0.041), and OSDI (*p* = 0.029) at 3 months. Greater continuous improvement in CFS (*p* = 0.045), dryness (*p* = 0.033), blurred vision (*p* = 0.031) and total symptom scores (*p* = 0.023) was demonstrated at 3 months than at 1 month post-treatment in the treatment group. We found that oral vitamin B1 and mecobalamin can improve corneal nerve length, width, reflectivity and the number of neuromas in IVCM, thereby repairing epithelial cells and alleviating some ocular symptoms. Thus, vitamin B1 and mecobalamin are potential treatment options for patients with DED.

## 1. Introduction

Dry eye disease (DED) is a prevalent ocular surface disease featured with loss of tear-film homeostasis and ocular symptoms, in which tear film instability, ocular surface inflammation and injury, and neurosensory abnormalities all play important etiological roles [1,2]. The TFOS DEWS II Pain and Sensation Subcommittee highlighted the neurobiological mechanisms that lead to discomfort accompanying DED in 2017 [3], and increasing evidence has demonstrated that in addition to providing sensation, corneal nerve structure and function play an important role in dry eye disease [3,4,5,6,7]. In recent years, with the increase in the understanding of DED and the progress of ocular surface assessment methods, in vivo confocal microscopy (IVCM) has proven to be useful to observe corneal microstructure in DED patients [3,4,5,6,7]. Patients with DED have observed alterations in subbasal corneal nerves in IVCM, including reduced density, high tortuosity, more branches and Langerhans cells (LC) [8,9,10,11,12,13,14,15,16,17], which correlates with DED symptoms, ocular surface staining [14,18], and the severity of dry eye [4,8,14]. A recent review proposed that corneal neuroma (sometimes referred to as micro neuromas), which may serve as a diagnostic biomarker, is a pathological feature of the ocular surface disease [19], and their presence may be related to the symptoms of photophobia and neuropathic ocular pain (NOP) [8,19]. In addition, the change in the density of corneal nerves during DED treatment plays an important role in the therapeutic response in patients with DED [11,18,20,21].

Vitamin B1 indirectly enhances axonal flow and transport, thus repairing damaged nerve tissue [22]. Vitamin B12 plays an important role in the regulation of neurotrophic factor synthesis, whose deficiency impairs sensory innervation [23,24,25]. Thiamine (vitamin B1) and mecobalamin (an endogenous form of vitamin B12) are widely used in treating pain, such as diabetic neuropathy and surgical pain [22,26,27,28]. A previous study found that treating glaucomatous patients with food supplements containing vitamin B1 could improve dry eye symptoms [29]. Several studies have shown that the parenteral and topical application of vitamin B12 could alleviate ocular pain and DED symptoms [25,30,31,32,33,34,35]. Topical vitamin B12 supplementation has been reported to reduce oxidative stress and inflammation [36,37], ameliorate both the morphology and function of corneal nerves [35,37], and improve corneal reinnervation and reepithelization after injury [38].

Currently, there are no case-control studies evaluating the effect of oral vitamin B1 and mecobalamin on corneal nerve changes in DED patients. Our previous study found that oral vitamin B1 and mecobalamin not only improved dry eye symptoms (especially dryness, pain, and photophobia), signs and patient satisfaction but also obviously increased corneal nerve fiber density (CNFD) [20]. Therefore, based on previous research, the present study with larger sample size, longer follow-up period and more observation parameters aimed to investigate the effects of oral vitamin B1 and mecobalamin on corneal nerves in DED patients, including the corneal nerve length, width, reflectivity and number of neuromas.

## 2. Methods

### 2.1. Ethical Considerations

The randomized, double-blind controlled study was performed at Peking University Third Hospital Eye Center from November 2018 to March 2021. The research was approved by the Human Research and Ethics Committee of Peking University Third Hospital (approval number M2018149) and adhered to the Declaration of Helsinki. Written informed consent was obtained from each patient before enrollment. The trial is registered at ClinicalTrials.gov (ChiCTR1900025047).

### 2.2. Participants

Patients diagnosed with DED at the Peking University Third Hospital Eye Center and aged 18 to 75 years were enrolled. DED was diagnosed based on the criteria established by the Tear Film and Ocular Surface Society (TFOS) Dry Eye Workshop II in 2017 [1]. Exclusion criteria consisted: (a) patients with diseases or received surgeries affecting corneal nerves, such as diabetes mellitus, trigeminal nerve injury, lagophthalmos, facioplegia, neurotrophic keratitis, history of corneal refractive surgery; (b) patients with active ocular surface inflammatory diseases unrelated to dry eye within six months; (c) other severe corneal diseases, glaucoma, optic neuropathy, uveitis, diabetic retinopathy; (d) complex systemic disease, such as Sjogren’s syndrome, Stevens-Johnson syndrome, and rheumatism; (e) patients with recent ocular surgery or contact lens wear within three months; (f) patients who currently received DED treatment (other than artificial tears), such as punctual plugs, thermal treatment; (g) patients who were allergic to oral vitamin B1 and mecobalamin; and (h) who were incapable of carrying out study-related visits.

### 2.3. Experimental Design

Eligible patients were randomly assigned 1:1 utilizing an automated Web-based randomization system to either the treatment group or the control group. Patients in the treatment group received oral vitamin B1 (Vitamin B1 Tablets, Shanxi Hengruida Pharmaceutical Co., Shanxi, China) and mecobalamin (mecobalamin tablets, Weicai [China] Pharmaceutical Co., Shenyang, China) one tablet three times a day and artificial tears (HYCOSAN, URSAPHARM Arzneimittel GmbH, Saarbrücken, Germany) five times daily. However, patients in the control group received artificial tears (HYCOSAN, URSAPHARM Arzneimittel GmbH, Saarbrücken, Germany) five times daily alone. Each patient underwent consecutive treatment for three months and received clinical assessment before treatment (baseline) and at one- and three-months post-treatment.

### 2.4. Evaluation Index

Before and at every follow-up post-treatment, each patient was asked to complete a questionnaire regarding subjective symptoms. Next, slit lamp examination and other auxiliary objective screenings were conducted to evaluate the DED signs. A 5 min interval was required between different tests. All ophthalmologic examinations were performed under the same conditions by a single investigator (Xiaotong Ren, an experienced resident). All symptoms and signs at each time point were compared within and between groups to evaluate the effects.

#### 2.4.1. Symptom Evaluation

All patients completed the same two questionnaires to evaluate DED symptoms before, at one month, and three months after treatment. First, the Ocular Surface Disease Index (OSDI) is scored from 0 to 100. The patients were required to answer 12 questions and give these items a score of 0 to 4. OSDI = (sum of scores for all questions answered × 100)/(total number of answered questions × 4). The second questionnaire comprised 10 questions related to 10 DED symptoms, whose responses ranged from “not present (score 0)” to “very serious (score 10)” and were rated on a 10-point scale. The total score was calculated as a sum of these 10 scores to evaluate the overall DED symptoms of patients [39].

#### 2.4.2. Sign Evaluation

We used a Placido ring-based, noninvasive ocular analyzer (Keratograph 5 M; OCULUS, Wetzlar, Germany) to assess the tear film, including the tear meniscus height (TMH) and tear film break-up time (TBUT). The device captured an infrared photograph of the anterior segment to evaluate TBUT and inferior TMH, which had proven the repeatability and reproducibility of keratography measurements [40]. The examination was performed before the slit-lamp examination and was repeated three times in an individual patient. Patients were required to blink 3 to 4 times before the examination and keep their eyes open as long as possible. TMH is the length of a vertical line from the top of the inferior tear meniscus to the eyelid margin and could be calculated automatically with the machine.

A slit lamp examination was performed to assess corneal fluorescein staining and conjunctival hyperemia. We performed the corneal staining by applying fluorescein sodium and then by a slit lamp with cobalt blue illumination. The cornea was divided into four quadrants. Staining scored 0 to 3 in each quadrant and then summed. The conjunctiva congestion score (CCS) was divided into four grades according to the Cornea and Contact Lens Research Unit (CCLRU) grading standard [41]: Grade 1 (0 points): no conjunctiva congestion. Grade 2 (1 point): mild conjunctival hyperemia confined to the fornix, with bright red blood vessels. In addition, normal blepharon texture; Grade 3 (2 points): moderate conjunctiva hyperemia, from the hyperemia site to the eyelid fissure, with deep red blood vessels and unclear vascular sites; Grade 4 (3 points): diffuse hyperemia of the conjunctiva, purplish red hyperemia of the blood vessels, and indistinct texture of the normal meibomian glands.

IVCM (HRT II RCM Heidelberg Engineering Inc., Heidelberg, Germany, Rostock Cornea Module) is a noninvasive imaging device to observe different layers of the cornea in vivo. Topical anesthesia was applied with oxybuprocaine before the examination, and hydroxypropyl methylcellulose was filled on the tip of the TomoCap for high image quality. Patients were required to fixate on a target, and we gradually moved the machine so that the IVCM could examine cornea of different depths. The images had 384 × 384 pixels definition over an area of 400 µm × 400 µm, a lateral spatial resolution of 0.5 µm and a depth resolution of 1–2 µm were captured. For each eye, 30 to 40 pictures were captured from the anterior to the posterior surface. Unclear images or images with artifacts were excluded. An experienced investigator (RXT) screened the pictures and chose five images with the richest plexus. Corneal nerve quantification was performed by another investigator (CYL). We applied ImageJ to analyze the width, length, and reflectivity of the subbasal nerves, and we chose the image with the maximum number of neuromas and reported the number per image [8]. Data were recorded as the average of 3 measurements.

### 2.5. Statistical Analysis

SPSS software version 23 (SPSS Inc., Chicago, IL, USA) was applied for statistical analysis. The Kolmogorov-Smirnov test was utilized to analyze the normality of the data distribution. The descriptive data were presented as the mean and standard deviation (SD). Considering the repeated measurements for a certain patient, we established a mixed linear model to compare differences pre- and post-treatment, in which the eye was chosen as the subject, the time point was the repeated factor, and the covariance type was chosen based on the covariance matrix from the preliminary analysis. The change at one or three months was calculated by subtracting the baseline measurement from the outcomes at one or three months. The T-test was adopted to compare the differences between the two groups for continuous variables, and the Wilcoxon nonparametric test was applied for ordinal variables. *p* values less than 0.05 were denoted statistically significant.

## 3. Results

### 3.1. Patients’ General Information

In this case, 398 eyes of 199 subjects who met the criteria of this study were enrolled. All subjects in each group completed the treatment and each follow-up visit. There were no significant differences between the two groups regarding age, sex, or other baseline signs and symptoms. The basic information about the patients is shown in Table 1.

### 3.2. The Effectiveness of the Treatment in Each Group

As shown in Table 2, in the treatment group, the conjunctival congestion score, symptoms of dryness, pain, photophobia, blurred vision, total symptom score, and OSDI were significantly improved between baseline and 1/3 months post-treatment (*p* < 0.05), and the corneal fluorescein staining (CFS) sign and asthenopia symptoms only improved significantly between baseline and three months post-treatment (*p* < 0.05).

In the control group, only the scores of asthenopia, photophobia, the total symptom score, and OSDI changed significantly one month after treatment (*p* < 0.05), but the dryness symptom differed significantly between one- and three-months post-treatment and between baseline and three months post-treatment (*p* < 0.05).

### 3.3. Comparison of the Effectiveness between Two Groups

To compare before and after treatment in the two groups (Table 3), patients who received vitamin B1 and mecobalamin showed greater improvement in the conjunctival congestion score (*p* = 0.036), dryness scores (*p* = 0.022) at one month, CFS (*p* = 0.012), blurred vision (*p* = 0.016), photophobia (*p* = 0.032), total symptom scores (*p* = 0.041), and OSDI (*p* = 0.029) at three months. Greater continuous improvement in CFS (*p* = 0.045), dryness (*p* = 0.033), blurred vision (*p* = 0.031) and total symptom scores (*p* = 0.023) was demonstrated at three months than at one-month post-treatment in the treatment group.

### 3.4. Changes in Corneal Subbasal Nerve Parameters in Each Group

In the treatment group, as shown in Table 4, there were significant improvements in corneal nerve length, width, and neuromas at 1/3 months and in corneal length between 1- and 3-months post-treatment (*p* < 0.05). In the control group, none of the parameters were significantly different during follow-up.

## 4. Discussion

This study proposed a new oral vitamin B1 and mecobalamin approach to treat DED. In this study, we found that oral vitamin B1 and mecobalamin could improve corneal nerve length, width, and neuromas in DED patients and ameliorated some symptoms and signs, including pain, photophobia, corneal fluorescein staining and conjunctival hyperemia. These observations indicate that oral vitamin B1 and mecobalamin facilitate nourishment and repair of the corneal nerve layer, thereby repairing epithelial cells and alleviating ocular symptoms.

The cornea has a nerve density of 300 to 600 times that of the skin and is the most densely innervated tissue [6,18,42]. Increasing evidence has demonstrated that corneal innervation plays an important role in ocular surface homeostasis and diseases in addition to providing sensation [43]. Alternation in the corneal nerve plexus can alter the complicated regulation of the ocular surface morpho-functional unit, which is associated with various functions, such as tear reflex, blinking, and trophism of the epithelial cells, which contributes to the dry eye vicious cycle [3,44]. The various pathogenic and clinical elements could necessitate different treatments [45]. Corneal nerve damage can lead to acute axonal injury and the release of inflammatory mediators such as interleukin-1, substance P, and tumor necrosis factor-α, thus reducing the ion channels threshold potential in corneal nerve endings, resulting in an aggravated corneal nociceptor response, which might result in symptoms including pain, burning, and photophobia [46]. The dry eye severity depends on the structural changes and function of the corneal nerve, and greater severity of dry eye is correlated with a larger number of neuromas and larger corneal nerve width on IVCM [8,14,16,19]. A study indicated that the length and density of corneal nerves were negatively correlated with DED symptoms and OSDI [14], and these indexes are the main quantifying parameter for corneal nerves identified with IVCM [9,16]. Hence, in our study, it is noteworthy that at 1 and 3 months after oral vitamin B1 and mecobalamin, DED patients responded well to an improvement in IVCM results, including corneal nerve length, width, and neuromas. 

Vitamin B1 and B12 are important nutrients for metabolic functions, whose deficiency may cause optic neuropathy and corneal epitheliopathy with decreased vision and photophobia [25,29,30,47,48,49]. They are widely used to treat pain, such as diabetic neuropathy and surgery-related pain, via indirect axonal transport to repair damaged nerve tissue [22,26,27,29,35,37,38,50,51]. Mecobalamin is one of the coenzyme forms of vitamin B12, and evidence indicates that this form of vitamin B12 has several metabolic and therapeutic functions different from other forms of vitamin B12 [23,24]. Our previous study found that oral vitamin B1 and mecobalamin could improve corneal nerve fiber density [20]. Paolo et al. proved that local vitamin B12 improved both the morphology and function of corneal nerves in diabetic patients [35], and Yang et al. found that nebulization with vitamin B12 could improve corneal nerve density [37]. In animal experiments, vitamin B12 could improve corneal reinnervation and re-epithelialization after injury [38]. In agreement with these findings, our study revealed a consecutive increase in subbasal nerve length at one month and three months after oral vitamin B1 and mecobalamin. The results might be explained by the evidence that vitamin B12 promotes β-III tubulin expression in neurons [52] and regulates neurotrophic factor synthesis, including upregulated TNF-α and downregulated neurotrophic epidermal growth factor (NEGF) [53], which contribute to neurite outgrowth and survival and promote epithelial wound healing. Vitamin B1 (thiamine) could also enhance the axonal flow between junctions [22]. Macri et al. indicated that vitamin B12 eye drops could reduce oxidative stress and inflammation [36]. They all support the improvement of CFS and CCS in the treatment group.

It is suggested that corneal fluorescein staining may be considered a probe for apoptotic epithelial cells [54,55]. Vitamin B12 possibly facilitates cellular energy production and nerve repair to restore the corneal nerves and epithelial cells. The present research demonstrated that the CFS improved at three months, which indicated that the repair process was long. A recent study demonstrated that using calf blood-deproteinized extract ophthalmic gel could improve conjunctival hyperemia and repair damaged corneal tissue and cells after two weeks [56]. Similarly, vitamin B 12 is crucial for DNA synthesis and cellular energy production [22,27,38], which might contribute to the CCS improvement in the treatment group at 1/3 months. Some studies have proven that the severity of conjunctival congestion is correlated to the severity of DED symptoms and signs, including eye pain, light sensitivity and CFS [57,58]. It is consistent with our results that oral vitamin B1 and mecobalamin showed greater improvement in CCS, CFS, photophobia and the total symptom scores compared with the control group.

In chronic DED, chronic neuropathic ocular pain (NOP) is observed, and changes occur in the ocular sensory apparatus. Chronic inflammation and nerve injury can result in abnormal activation of the ocular sensory fibers, leading to neuropathic pain [3,7]. Vitamin B1 (thiamine) administered perineurally has analgesic effects by promoting the synthesis of acetylcholine in the inhibitory neurons of the dorsal horn [22]. Serkan et al. [25] found that vitamin B12 deficiency is related to neuropathic ocular pain (NOP), and parenteral vitamin B12 supplementation could improve resistant ocular pain in patients with severe DED after 12 weeks. Another case also reported that a patient treated with parenteral vitamin B12 was pain-free and completely relieved from all symptoms [29]. In a review, mecobalamin demonstrates potential analgesic effects on neuropathic pain by inhibiting spontaneous ectopic discharges from peripheral primary sensory neurons and improvements in injured nerve regeneration in peripheral hyperalgesia and allodynia [50]. Our previous study found that oral vitamin B1 and mecobalamin could improve dry eye symptoms at two months, especially dryness, pain, and photophobia [20]. Similarly, the current study indicated that oral vitamin B1 and mecobalamin could improve pain, photophobia and blurred vision symptoms, the total symptom score and the OSDI at 1/3 months. Ocular pain correlates with symptom severity and persistence [59], which may contribute to the improvement of total symptom score and OSDI in the treatment group, and it explains why the scores of asthenopia, photophobia, the total symptom score, and OSDI changed significantly only one month after treatment in the control group. At three months post-treatment, the control group received artificial simply and only differed significantly in dryness symptoms. Previous studies indicated that treatments with topical artificial tears are insufficient for pain improvement in many DED patients [25], and the unsatisfied improvement in DED symptoms are associated with NOP [60].

The meibum secreted by the meibomian gland (MG) could reduce tear evaporation, so meibomian gland function is crucial for tear film stability. TBUT is an important indicator of meibomian gland function. Unfortunately, we did not observe an improvement in BUT in this study or our previous research [20], which was different from some previous studies [34,36,37]. I think it probably because the different forms of administration, and the local administrations could obviously improve TBUT. However, there was no morphological and functional analysis of meibomian gland ducts in previous studies, so the separate mechanism of the meibomian gland is not clear. We will further assess Meibomian gland function (secretion quality, expressibility and so on), morphology of MG ducts, and eyelid margin abnormalities to explore their effect and mechanism on Meibomian gland dysfunction in future studies. 

In summary, our study suggests a positive effect of oral vitamin B1 and mecobalamin in ameliorating both corneal nerve length, width, and neuromas in DED patients, improving some symptoms and signs, such as pain, photophobia, corneal fluorescein staining and conjunctival hyperemia. We speculate that neurotrophism might be the underlying mechanism. Our findings raise several possible fields of study for oral vitamin B1 and mecobalamin, such as the therapeutic effects on other diseases in which corneal nerves are acutely injured (refractive and corneal surgery) or the application on patients complaining of ocular pain or severe DED with abnormal nerve regulation. A limitation of this study is the relatively short follow-up period. In addition, we did not evaluate corneal sensitivity. Our subsequent research will take longer follow-ups and evaluate the possible efficacy of oral vitamin B1 and mecobalamin in different populations (diverse types of DED; moderate-severe DED; low or absent nerve damage; different levels of serum vitamin B1 and vitamin B12). In addition, the dose and frequency of treatment could be investigated to determine an optimal therapeutic regimen. The key mechanisms of ocular nebulization therapy on DED require further investigation.

## Figures and Tables

**Table 1 nutrients-14-03750-t001:** Patients’ general information.

Signs and symptoms	Items	Control Group	Treatment Group	*p*
	Number of eyes	210	188	
	Sex (M/F)	35/70	31/63	ns
	Age (Years)	52.58 ± 18.02	53.94 ± 17.22	ns
Sign	TMH (mm)	0.19 ± 0.01	0.18 ± 0.01	ns
	TBUT (s)	5.40 ± 0.22	5.38 ± 0.21	ns
	CFS	0.29 ± 0.09	0.58 ± 0.13	ns
	CCS	0.09 ± 0.02	0.11 ± 0.02	ns
Symptoms (Scores)	Dryness	6.14 ± 0.23	6.31 ± 0.26	ns
	Foreign body sensation	4.52 ± 0.27	4.70 ± 0.30	ns
	Pain	3.38 ± 0.27	3.89 ± 0.28	ns
	Burning	2.22 ± 0.25	2.30 ± 0.26	ns
	Watering	3.17 ± 0.27	3.10 ± 0.30	ns
	Asthenopia	5.88 ± 0.27	5.44 ± 0.30	ns
	Blurred vision	3.82 ± 0.30	3.20 ± 0.29	ns
	Itching	3.41 ± 0.26	2.91 ± 0.29	ns
	Increased secretions	3.12 ± 0.25	3.58 ± 0.31	ns
	Photophobia	4.54 ± 0.29	4.38 ± 0.31	ns
	Total	40.09 ± 1.60	39.55 ± 1.93	ns
	OSDI	41.92 ± 2.49	39.37 ± 2.46	ns

M: male; F: female; TMH: tear meniscus height; TBUT: tear film break-up time first; CFS: corneal fluorescein staining, CCS: conjunctival congestion score; ns: no statistical significance.

**Table 2 nutrients-14-03750-t002:** Comparison of pre-and post-treatment in each group.

Signs and Symptoms		Control Group			Treatment Group		
		B	1	3	B	1	3
Signs (mean ± SD)	TMH (mm)	0.19 ± 0.01	0.20 ± 0.01	0.19 ± 0.01	0.18 ± 0.01	0.18 ± 0.01	0.19 ± 0.01
	TBUT(s)	5.40 ± 0.22	5.72 ± 0.26	5.51 ± 0.48	5.38 ± 0.21	4.98 ± 0.25	5.68 ± 0.35
	CFS	0.29 ± 0.09	0.03 ± 0.10	0.16 ± 0.12	0.58 ± 0.13	0.21 ± 0.16	0.12 ± 0.17 ^#^
	CCS	0.09 ± 0.02	0.07 ± 0.02	0.05 ± 0.03	0.11 ± 0.02	0.01 ± 0.03 *	0.05 ± 0.03 ^#^
Symptoms (mean ± SD, scores)	Dryness	6.14 ± 0.23	5.98 ± 0.26	5.18 ± 0.31 ^𝝏#^	6.31 ± 0.26	5.23 ± 0.31 *	5.16 ± 0.33 ^#^
	Foreign body sensation	4.52 ± 0.27	3.77 ± 0.29	4.04 ± 0.35	4.70 ± 0.30	4.12 ± 0.36	3.75 ± 0.39
	Pain	3.38 ± 0.27	2.75 ± 0.30	3.30 ± 0.37	3.89 ± 0.28	2.51 ± 0.32 *	2.99 ± 0.34 ^#^
	Burning	2.22 ± 0.25	2.27 ± 0.27	1.87 ± 0.37	2.30 ± 0.26	1.72 ± 0.29	1.90 ± 0.31
	Watering	3.17 ± 0.27	2.69 ± 0.29	2.45 ± 0.35	3.10 ± 0.30	2.81 ± 0.34	2.59 ± 0.36
	Asthenopia	5.88 ± 0.27	4.99 ± 0.30 *	5.38 ± 0.37	5.44 ± 0.30	5.09 ± 0.34	4.49 ± 0.37 ^#^
	Blurred vision	3.82 ± 0.30	3.12 ± 0.32	3.61 ± 0.39	3.20 ± 0.29	2.38 ± 0.34 *	2.49 ± 0.36 ^#^
	Itching	3.41 ± 0.26	2.98 ± 0.28	3.14 ± 0.34	2.91 ± 0.29	2.89 ± 0.33	2.84 ± 0.35
	Increased secretions	3.12 ± 0.25	2.97 ± 0.28	3.59 ± 0.35	3.58 ± 0.31	3.00 ± 0.35	2.27 ± 0.37
	Photophobia	4.54 ± 0.29	3.93 ± 0.32 *	3.97 ± 0.38	4.38 ± 0.31	3.95 ± 0.35 *	3.66 ± 0.38 ^#^
	Total	40.09 ± 1.60	34.93 ± 1.72 *	36.18 ± 2.03	39.55 ± 1.93	33.98 ± 2.15 *	32.19 ± 2.28 ^#^
	OSDI	41.92 ± 2.49	34.44 ± 2.54 *	35.32 ± 2.98	39.37 ± 2.46	33.99 ± 2.69 *	29.34 ± 2.82 ^#^

B: baseline; 1: 1 month; 3: 3 months; TMH: tear meniscus height; TBUT: tear film break-up time; CFS: corneal fluorescein staining, CCS: conjunctival congestion score. *p* values of less than 0.05 were considered statistically significant and are expressed as * (1 month vs. baseline), ^#^ (3 months vs. baseline), and ^𝝏^ (3 months vs. 1 month) in the same group.

**Table 3 nutrients-14-03750-t003:** Comparison of pre-and post-treatment between the two groups.

Signs and Symptoms		1 Month—Baseline			3 Months—Baseline			3 Months–1 Month		
		Control Group	Treatment Group	*p*	Control Group	Treatment Group	*p*	Control Group	Treatment Group	*p*
Signs (mean ± SD)	TMH (mm)	0.009 ± 0.006	0.005 ± 0.005		0.002 ± 0.011	0.010 ± 0.007		0.008 ± 0.011	0.004 ± 0.008	
	TBUT (s)	0.322 ± 0.275	−0.400 ± 0.299		0.108 ± 0.487	0.399 ± 0.388		0.214 ± 0.502	0.699 ± 0.410	
	CFS	−0.265 ± 0.104	−0.377 ± 0.174		−0.136 ± 0.130	−0.366 ± 0.182	0.012	0.129 ± 0.137	−0.111 ± 0.203	0.045
	CCS	−0.026 ± 0.027	−0.096 ± 0.036	0.036	−0.040 ± 0.034	−0.055 ± 0.038		−0.036 ± 0.035	−0.051 ± 0.041	
Symptoms (mean ± SD, scores)	Dryness	−0.153 ± 0.230	−1.085 ± 0.308	0.022	−0.958 ± 0.294	−1.153 ± 0.337		−0.804 ± 0.307	−1.168 ± 0.372	0.033
	Foreign body sensation	−0.748 ± 0.249	−0.582 ± 0.362		−0.480 ± 0.323	−0.948 ± 0.396		0.268 ± 0.338	−0.366 ± 0.437	
	Pain	−0.627 ± 0.279	−1.377 ± 0.291		−0.082 ± 0.360	−0.898 ± 0.316		0.545 ± 0.377	0.479 ± 0.349	
	Burning	0.051 ± 0.245	−0.574 ± 0.259		−0.353 ± 0.318	−0.400 ± 0.285		−0.404 ± 0.333	0.174 ± 0.315	
	Watering	−0.782 ± 0.248	−0.295 ± 0.264		−0.715 ± 0.323	−0.507 ± 0.291		0.067 ± 0.337	−0.212 ± 0.322	
	Asthenopia	−0.889 ± 0.277	−0.347 ± 0.317		−0.503 ± 0.358	−0.566 ± 0.341		−0.386 ± 0.375	−0.805 ± 0.215	
	Blurred vision	−0.699 ± 0.276	−0.823 ± 0.317		−0.213 ± 0.358	−0.719 ± 0.348	0.016	−0.485 ± 0.375	−0.705 ± 0.384	0.031
	Itching	−0.433 ± 0.229	−0.020 ± 0.295		−0.266 ± 0.299	−0.070 ± 0.320		−0.167 ± 0.312	−0.050 ± 0.354	
	Increased secretions	−0.223 ± 0.268	−0.579 ± 0.308		0.402 ± 0.345	−1.312 ± 0.339		0.625 ± 0.362	−0.733 ± 0.375	
	Photophobia	−0.512 ± 0.269	−0.433 ± 0.298		−0.369 ± 0.350	−0.725 ± 0.328	0.032	−0.243 ± 0.366	−0.392 ± 0.363	
	Total	−5.158 ± 1.365	−5.577 ± 1.731		−4.903 ± 1.768	−7.360 ± 1.908	0.041	−3.255 ± 1.852	−7.784 ± 2.112	0.023
	OSDI	−5.380 ± 2.300	−5.380 ± 2.300		−6.598 ± 3.315	−10.969 ± 2.560	0.029	1.119 ± 3.396	−4.349 ± 2.777	

TMH: tear meniscus height; TBUT: tear film break-up time first; CFS: corneal fluorescein staining, CCS: conjunctival congestion score. *p* values of less than 0.05 were considered statistically significant.

**Table 4 nutrients-14-03750-t004:** Changes in corneal subbasal nerve parameters in each group.

	Control Group			Treatment Group		
	B	1	3	B	1	3
Corneal length (pixels)	2680.86 ± 773.94	2624.41 ± 663.03	2721.93 ± 634.02	2667.21 ± 651.44	2738.35 ± 617.43 *	3081.27 ± 626.50 ^𝝏#^
Corneal width (pixels)	5.86 ± 1.11	6.15 ± 0.98	5.83 ± 0.0.76	5.43 ± 1.31	4.96 ± 0.86 *	4.88 ± 0.67 ^#^
Corneal reflectivity (gray values)	142.46 ± 20.29	150.83 ± 18.43	146.53 ± 17.84	132.41 ± 26.68	149.53 ± 21.21	155.44 ± 16.39 ^#^
Corneal neuromas (N)	0.80 ± 0.70	0.78 ± 0.67	0.80 ± 0.52	0.81 ± 0.75	0.71 ± 0.61 *	0.56 ± 0.51 ^#^

B: baseline; 1: 1 month; 3: 3 months. *p* values of less than 0.05 were considered statistically significant and are expressed as * (1 month vs. baseline), ^#^ (3 months vs. baseline), and ^𝝏^ (3 months vs. 1 month) in the same group.

## Data Availability

The datasets generated and analyzed during the current study are not publicly available but are available from the corresponding author on reasonable request.

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
