# Peer review of "The Utility of Oral Vitamin B1 and Mecobalamin to Improve Corneal Nerves in Dry Eye Disease: An In Vivo Confocal Microscopy Study"

_nutrients, 2022, doi:10.3390/nu14183750_

Round 1

Reviewer 1 Report

I thank the authors for the opportunity to review their study.

Overall this is a thoughtful paper that considers and discusses possible mechanism of action of oral Vitamin B1 and Mecobalamin very well.

My general impressions of the study design and interpretation are positive. However, one thing that struck me as I began reading was the some of the sentences presented are so close to previously published reports and to be considered plagiarism. The authors are exhorted to return to the manuscript and make sure they are not infringing upon the original work of others in presenting almost verbatim the words of previous researchers.

Overall the language needs to be revisited- The plagiarism is hopefully inadvertent but there are other instances in which the language is a bit clumsy.

More minor comments below:

Line 18: Cornea nerve change to Corneal

Line 24: Need to  write out CFS in full first time it is mentioned.

Line 44- “observed” is a better word here than “revealed”

What do you mean by more beadings Line 46?

Line 197: I  normally expect to see Control group to the left of Treatment group. And also the Table would be improve by having a column for P values or some other way making it easier for the reader to quickly identify the significant results rather than the small superscripted symbols currently used.

In relation to Total scores reported for signs and symptoms. Can the authors justify giving equal weight to each sign and symptom? They do offer a reference but that reference doesn’t justify this methodology either. How do we know that each sign and symptom contributes equally to the patient’s overall experience of dry eye?

Line 209: Again, I expect to see Control group to the left of Treatment group

Line 217: same comments for Table 4 as for Table 2

Lines 226-227: “Ameliorate” and “ameliorated” are used in quick succession. Consider changing one of these.

Line 227: What do the authors mean by “ameliorated the homeostasis of the ocular surface”? What aspects of the corneal surface are brought back into homeostasis? Homeostasis suggests to me that a new equilibrium has been achieved which would seem to indicate expectation of the new situation even after the treatment has been stopped. Can the authors comment on this?

Line 271: Change “et al” to “et al.” (And correct all other instances)

The discussion overall is very well written but is a bit long. Is it possible to make this section more concise?

Line 298: what do the authors mean by “This is nearly consistent”? Need to expand upon this further and be more detailed.

Reviewer 2 Report

This is an article entitled “The Utility of Oral Vitamin B1 and Mecobalamin to Improve 2 Corneal Nerves in Dry Eye Disease: An In Vivo Confocal Mi-3 croscopy Study (nutrients-1854798)” which evaluates the changes in cornea nerve parameters and symptoms and signs in dry eye disease patients after oral vitamin B1 and mecobalamin treatment.

Abstract

-         -  Good.

Introduction

-         - Good.

Methods

-         - Please also give the upper most age of the included subjects.

-          - Were the patients aquous or evaporative dry eye cases? Or were they mixed type? Did you check the MGD presence?

Results

-        -  Please give the ranges of all data.

Discussion

-          - Please also make a seperate paragraph to discuss the possible effects of these agents on Meibomian gland dysfunction as well.

References

-         -  Good.

Tables

-          - Please give the ranges of all data.
